

# Modelling African biomass burning emissions and the effect of spatial resolution

Dave van Wees[1], Guido R. van der Werf[1]

[1]Department of Earth Sciences, Vrije Universiteit, Amsterdam, 1081 HV, the Netherlands

*Correspondence to*: Dave van Wees (d.van.wees@vu.nl)

**Abstract.** Large-scale fire emission estimates may be influenced by the spatial resolution of the model and input datasets used. Especially in areas with relatively heterogeneous land cover, a coarse model resolution might lead to substantial errors in estimates. In this paper, we developed a model using Moderate Resolution Imaging Spectroradiometer (MODIS) satellite observations of burned area and vegetation characteristics to study the impact of spatial resolution on modelled fire emission

estimates. We estimated fire emissions for sub-Saharan Africa at 500-meter spatial resolution (native MODIS burned area) for the 2002-2017 period, using a simplified version of the Global Fire Emissions Database (GFED) modelling framework, and compared this to model runs at a range of coarser resolutions (0.050°, 0.125°, 0.250°). We estimated fire emissions of 0.68 PgC yr[-1] at 500-meter resolution and 0.82 PgC yr[-1] at 0.25° resolution; a difference of 24%. At 0.25° resolution, our model results were relatively similar to GFED4, which also runs at 0.25° resolution, whereas our 500-meter estimates were

substantially lower. We found that lower emissions at finer resolutions are mainly the result of reduced representation errors when comparing modelled estimates of fuel load and consumption to field measurements, as part of the model calibration. Additional errors stem from the model simulation at coarse resolution and lead to an additional 0.02 PgC yr[-1] difference in estimates. These errors exist due to the aggregation of quantitative and qualitative model input data; the average- or majority- aggregated values are propagated in the coarse resolution simulation and affect the model parameterization and the

final result. We identified at least three error mechanisms responsible for the differences in estimates between 500-meter and 0.25° resolution simulations, besides those stemming from representation errors in the calibration process, namely: 1. biome misclassification leading to errors in parameterization, 2. errors due to the averaging of input data and the associated reduction in variability, and 3. a temporal mechanism related to the aggregation of burned area in particular. Even though these mechanisms largely neutralized each other and only modestly affect estimates at a continental scale, they lead to

substantial error at regional scales with deviations up to a factor 4, and may affect large-scale estimates differently for other continents. These findings could prove valuable in improving coarse resolution models and suggest the need for increased spatial resolution in global fire emission models.



# 1 Introduction

Fires exert a key influence on the global climate by the release of trace gases and aerosols into the atmosphere (Andreae and Merlet, 2001; Ciais et al., 2013; Ward et al., 2012). Furthermore, fires partly shape, and in the long-term sometimes determine, the vegetation state of landscapes, thus affecting the storage capacity of carbon (Rabin et al., 2017). About 70%

of global burned area occurs in Africa (Giglio et al., 2018), leading to roughly half of the global fire carbon emissions (van der Werf et al., 2010). The majority of fires in Africa occur in the savannas (Archibald et al., 2009; van der Werf et al., 2017), an ecosystem that is dependent on fires and where trees have evolved to tolerate fire (Beerling and Osborne, 2006). African savannas are currently undergoing major shifts in fire activity due to demographic changes and agricultural expansion, leading to a decrease in fire occurrence (Andela and van der Werf, 2014).

Efforts to estimate global fire emissions have been made since the eighties (Seiler and Crutzen, 1980). Early estimates were based on biome-specific parameterizations of fire return times and biomass consumption rates, extrapolated using vegetation maps. More recently, satellite products have become an important tool for improved estimates of fire emissions, mapping fire events globally and giving insight in fire impacts and dynamics. Two main satellite-based approaches to estimate

emissions exist, based either on observed burned area in combination with a biogeochemical or fuel load model, or based on fire radiative power (FRP), which is directly related to fire emissions after integration over time to obtain fire radiative energy (FRE) (Kaiser et al., 2012; Wooster, 2002). Burned area is determined after a fire has occurred, signified by a change in surface reflectance associated with the burn scar (Giglio et al., 2018), whereas FRP is based on the fire size and intensity, determined by detection of the thermal hot spot during a satellite overpass.

In fire emission models, aboveground biomass and resulting fuel load are key variables for estimating emissions. Biogeochemical models dynamically simulate biomass buildup and degradation, and come with different levels of process complexity (Hély et al., 2003, 2007; Hoelzemann et al., 2004; Schultz et al., 2008; van der Werf et al., 2017). In regional models, parameterizations derived from field data can be used to accurately represent local relations between e.g.

precipitation and plant productivity, and between soil moisture and combustion completeness, and resulting fuel load can be calibrated at local scale (Alleaume et al., 2005; Hély et al., 2007; Korontzi et al., 2004; Russell-Smith et al., 2009). Some of these models are based on predetermined fuel load maps (Ito and Penner, 2004). However, in global-scale models, simple parameterizations are often inaccurate due to the large variety in e.g. vegetation dynamics and fire characteristics across continents and biomes (Lehmann et al., 2014; Rogers et al., 2015). As a result, these models often depend heavily on

satellite-derived climate and weather data, and land and vegetation characteristics. However, global satellite data on fire-specific processes is scarce (Pettinari and Chuvieco, 2016). Therefore, field measurements are crucial in constraining modelled fuel load and consumption (Hély et al., 2003; van Leeuwen et al., 2014). Modelled fuel load can be combined with combustion completeness factors to estimate fuel consumption, and then with satellite-based burned area maps to estimate



dry matter emissions. Finally, emission factors are used to convert dry matter or carbon emissions into emissions of trace gases and aerosols, which are key inputs for atmospheric and Earth system models (Akagi et al., 2011; Meyer et al., 2012; Wooster et al., 2011; Yokelson et al., 2013).

The detection of burn scars is limited by the spatial resolution of the satellite detector, as burned patches smaller than the satellite footprint are often not detected. When these relatively small fires are active during the satellite overpass, the thermal anomaly and its FRP may be detectable. Recent burned area products combine both of these detection methods to complement burned area based on burn scar detection with relatively small fires from active fire detection. In a first study looking into this on a global scale, Randerson et al. (2012) found an increase in global burned area of approximately 35%

due to the addition of small fire burned area. These small fires are often human-induced (prescribed, agricultural, deforestation) and mainly occur in croplands, woody savannas and tropical forests. Consequently, by the inclusion of these small fires, global fire emission estimates based on burned area from the Global Fire Emissions Database (GFED) increased from 1.5 PgC yr$^{-1}$ in GFED4 to 2.2 Pg C yr$^{-1}$ in GFED4s ("s" for small fires) on average over 1997-2016 (van der Werf et al., 2017). For sub-Saharan Africa alone, emissions increased from 0.8 Pg C yr$^{-1}$ in GFED4 to 1.1 Pg C yr$^{-1}$ in GFED4s.

Besides the error in burned area due to limitations of the satellite detector and undetected small fires (amongst other things), the accuracy of fire emission estimates may also be affected by the coarse spatial resolution of most fire emission models. Emission models based on burned area, such as GFED4, often perform at a spatial resolution significantly coarser than the native resolution of the burned area dataset. This is necessary because input data used to calculate emissions, especially

meteorological data, is usually much coarser than satellite data. Because of this and the necessary tradeoff between model complexity and computational resources, the burned area data is spatially aggregated to coarser resolution prior to the model simulation (e.g. 0.25° spatial resolution used in GFED4). However, there might be large heterogeneity of fuels and combustion characteristics within aggregated burned area (Alleaume et al., 2005; Hély et al., 2003). Whether aggregation, and the associated loss in heterogeneity, leads to significant errors in large-scale averaged model estimates such as GFED is

not known. Therefore, it is necessary to understand the implications of spatial aggregation for the accuracy of modelled fire emissions.

Previous studies have examined how relatively coarse spatial resolution could lead to biases in the results of remote sensing studies. For example, Eva & Lambin (1998) analyzed biases in 30 m Landsat TM burned area for Central Africa after spatial

aggregation to a resolution of 1 km. Similarly, García Lázaro et al. (2013) studied the burned area classification error in Iberia for several satellite products that span a range of resolutions (250m, 1100m, 0.05°), as compared to the 30m Landsat product. Comparable studies were done at continental scale by Silva et al. (2005) for Africa and Miettinen & Liew (2009) for Southeast Asia. All of the previously mentioned studies found that at coarser resolution, small and fragmented burned area tends to be underestimated compared to the finest resolution available data, whereas large fires and spatial homogeneity



leads to better estimates (with a tendency to overestimate). Nelson et al. (2009) specifically studied the impact of spatial aggregation, by comparing majority and average-based aggregation of an inventory-based forest classification (forest or non-forest). For majority-based aggregation, they reached conclusions analogous to the previously mentioned burned area studies, namely that at coarser resolution the forest proportion is underestimated for sparsely forested area, whereas it is

overestimated for heavily forested area. For average-based aggregation however, the mean forest proportion remained constant, as binary area is averaged to fractional area in the aggregate pixels. Furthermore, image variability decreased for coarser resolutions, because the average-aggregated pixel values converge towards the mean value of the entire image (Bian, 1997).

Errors introduced by spatially aggregating fine resolution input datasets to coarser resolution are propagated in the models driven by these datasets (Crosetto et al., 2001). When aggregated datasets are used in a nonlinear model, an additional error arises due to the nonlinear propagation of averaged values, known as Jensen's Inequality (Jensen, 1906). In general, for every nonlinear function there exists an inequality between taking the average of the function result afterwards, versus averaging the function input variables beforehand. We could, for example, consider a fire emission model as a single

nonlinear function. When running this model at aggregated resolution, an inequality (i.e. error) exists compared to the native resolution model. The magnitude of the inequality is dependent on the variance of, and covariance between, the input variables, and the amount of local curvature (second derivative) of the function, which is a measure of its non-linearity (Denny, 2017). Jensen's inequality is mostly discussed in literature in relation to ecology (Cale et al., 1983; Duursma and Robinson, 2003; Pierce and Running, 1995; Ruel and Ayres, 1999), but also in relation to biology (Denny, 2017) and

geology (Heuvelink and Pebesma, 1999), in the context of spatial, temporal and class averaging (e.g. plant functional types, PFTs). However, the implications of this inequality for fire emission estimates is not known. The resulting error in emission estimates could be of particular importance, since fire processes are generally highly heterogeneous (Randerson et al., 2012; Roy and Landmann, 2005).

In this context, the aim of this study is to better understand the impact of spatial resolution on the resulting biomass and fire emission estimates. Whether the aforementioned errors from modelling at aggregated resolutions result in significant errors in large-scale averaged fire emission estimates such as GFED, and fire adapted Dynamic Global Vegetation Models (DGVMs, e.g. those used in FireMIP) has until now not been investigated (Rabin et al., 2017; van der Werf et al., 2017). To this end, we developed a fire emission model capable of running at 500-meter spatial resolution, to produce a first emission

estimate at this resolution for sub-Saharan Africa. We then compared these emission estimates to three additional simulations using the same model for a range of aggregated resolutions (0.25°, 0.125°, 0.05°), in order to study the impact of spatial resolution on model results. Besides a comparison of large-scale emission estimates, a substantial part of our work was to understand local-scale biases due to aggregation, and identifying the underlying error mechanisms. As part of this analysis, we also considered the role of modelled biomass, a key precursor for resulting emissions. Finally, we compared our



500-meter and 0.25° resolution model results to the emission estimates from GFED4(s) (van der Werf et al., 2017), and tried to contextualize the changes in emission estimates due to modelling at aggregated resolutions in respect to changes due to model validation improvements and the incorporation of small fires. The insights gained in this study could possibly form an important step forward in the direction of global fire emission modelling at native satellite resolution, and in implementing

counter-measures for reducing errors when modelling at aggregated resolutions.

## 2. Methods

We developed a model to estimate fire emissions for sub-Saharan Africa for the 2002-2017 period with a monthly time step. We start with describing the model, which was derived from the GFED modelling framework and adapted to run at a range of spatial resolutions (2.1). This is followed by a description of the various input datasets (2.2). We then describe the model

optimization using satellite-based reference data and field measurements of fuel load (FL) and fuel consumption (FC) (2.3). Finally, we describe the simulations performed (2.4) and the methods used to compare different model resolutions (2.5).

### 2.1 Model description

For this study a simplified version of the GFED model was used. GFED is rooted in the Carnegie-Ames-Stanford-Approach (CASA) biosphere model, which was developed to simulate the terrestrial carbon cycle using satellite data to constrain

carbon uptake and other fluxes (Field et al., 1995; Potter et al., 1993). Van der Werf et al. (2003) extended this model to include fire processes, and provided spatially resolved estimates of fire emissions for the (sub)tropics. Over time, further modifications were made to GFED, including improved burned area identification (Giglio et al., 2006, 2013) and distinction between different sources of fire emissions on a global scale (van der Werf et al., 2006, 2010). The most recent version, GFED4s, also aims to account for relatively small fires that remain undetected by most burned area algorithms (Randerson et

al., 2012; van der Werf et al., 2017). These small fires add about 15% burned area in our study area in Africa. Recent research suggests this increase in burned area may be conservative (Roteta et al., 2019). For this study we have simplified the GFED model so it can be run at 500-meter resolution on continental scale; as compared to the 0.25° resolution of GFED4s running on a global scale. Only the main GFED functionality relevant for aboveground dynamics in biomass, litter and fire emissions was maintained. More refined mechanisms represented in GFED, such as belowground dynamics,

herbivory, grazing and fuelwood collection were not implemented. Furthermore, no specific deforestation mechanisms were modelled. These simplifications not only made required computational resources manageable, but also made it easier to disentangle mechanisms that cause differences between the model runs at different resolutions, which was our key objective.

The model has a pool-based structure wherein Net Primary Productivity (NPP) is partitioned over various biomass pools,

that are affected by losses due to turnover and fire processes. Aboveground biomass (AGB) and belowground biomass (BGB) are considered as the live part of the total available carbon above and below the ground, and the total aboveground



live and dead carbon is referred to as aboveground biomass and litter (AGBL), all expressed in mass of carbon per unit area (g C m$^{-2}$). NPP was calculated as the product of incoming solar radiation (SSR), the fraction of photosynthetically active radiation (fPAR) and a biome-specific light-use efficiency (LUE; $\varepsilon_{biome}$):

$$NPP(x,t) = SSR(x,t) \cdot fPAR(x,t) \cdot \varepsilon_{biome} ,$$ (1)

where $x$ is the grid location coordinate and $t$ is the time in months. NPP was distributed over tree and non-tree vegetation classes by multiplication with fractions of tree and non-tree vegetation cover, and further distributed in equal parts over the corresponding biomass pools. Trees were represented as leaf, stem and root pools, all receiving one-third of tree allocated NPP. Non-tree vegetation was represented as grass and root pools, both receiving half of non-tree allocated NPP. In this simplified categorization other non-tree vegetation types, such as shrubs, are part of the grass pool. For trees the root pool was subdivided into separate fine and coarse root pools, with 20% of the stem NPP allocated to the coarse roots, whereas for non-tree vegetation all root biomass consisted of fine roots. We used biome-specific LUE values based on those reported by Field et al. (1995). Since LUE was not reported for the savanna biome, we used the open shrubland value of 0.208 gC/MJ for open savannas, and an empirically determined value of 0.280 gC/MJ for woody savannas (see also Table 1). The LUE value for woody savannas was chosen to be in-between values reported for forest and grassland biomes.

When the model reaches its equilibrium state after the spin up phase, the carbon input from NPP is balanced by the carbon output via fires and respiration because of decomposition. Depending on pool-specific turnover rates and fire processes, biomass decays into three litter pools: fine litter, coarse woody debris (cwd), and soil organic matter. The pool-specific turnover rates, loosely based on those used in GFED4 (van der Werf et al., 2017), were optimized to biome-specific values in a series of model validation steps (see section 2.3, 'Model optimization'). The vegetation exposed to fire is either combusted and emitted as carbon directly, killed and converted to litter, or unaffected by the fire. The amount of biomass and litter exposed to fire was calculated by multiplication of the available flammable carbon and the burned fraction for each pixel:

$$C(x,t) = \sum_{\#\,pools}\left[AGBL_{pool}(x,t) \cdot M_{tree}(FTC) \cdot CC_{pool}\left(\varepsilon_{SM}(x,t)\right)\right] \cdot BA(x,t) \cdot f_c ,$$ (2)

where $C$ is the amount of carbon combusted and released to the atmosphere, $M_{tree}$ is a fire-induced tree mortality scalar, $CC$ is the combustion completeness, $BA$ is the burned area, i.e. fraction of pixel burned, $f_c$ is the fraction of carbon in fuel, for which we used 50%. The part of fire-exposed carbon that is combusted was determined by pool-specific combustion completeness values that were scaled linearly between a predefined minimum and maximum value dependent on an empirically defined soil moisture scalar. This scalar was defined as:

$$\varepsilon_{SM}(x,t) = \frac{\frac{SM(x,t)}{0.37}-0.4}{0.6} \quad with \;\; 0.1 < \varepsilon_{SM} < 1.0 ,$$ (3)

where $SM$ is the volumetric soil water content in units of volume fraction. The scalar was obtained by first standardizing the $SM$ values to a range between 0 and 1, and then dividing by 0.6 and capping at 1 to remove anomalously high values related





to wetlands. Additionally, the scalar values were capped to not be lower than 0.1, simulating a minimum soil moisture level below which moisture dependent processes are not further affected. Dry conditions result in CC values closer to the maximum, and vice versa for wet conditions. A mortality scalar for woody vegetation simulated whether trees exposed to fire are killed, and consequently directly combusted, or left as litter (van der Werf et al., 2003). This scalar was expressed as

the squared fraction of tree cover to total vegetation, to resemble the range from low fire-induced mortality in open landscapes (where trees are adapted to fire) to high mortality in dense tropical forests where trees are not adapted. When a tree is killed, all of unburned aboveground and belowground biomass is transferred to the litter pools. More specifically, leaves and grass become fine litter, dead stems are added to the cwd pool, and dead roots are added to the soil pool.

The decomposition of litter is dependent on temperature and moisture conditions. The rate of decomposition was based on pool-specific turnover rates, and scaled by an abiotic scalar. The abiotic scalar ($\varepsilon_A$) was defined as:

$$\varepsilon_A = \frac{\varepsilon_T \varepsilon_{SM}}{0.9} \quad with \quad 0.1 < \varepsilon_A < 1.0 \, , \tag{4}$$

where $\varepsilon_T$ is the temperature scalar:

$$\varepsilon_T = Q_{10}^{\frac{T-30}{10}} \quad with \quad \varepsilon_T > 1.0 = 1.0 \, , \tag{5}$$

where $T$ is the temperature in °C, and $Q_{10}$ is the temperature coefficient, for which we used a value of 1.5, and a capped maximum of 1.0 at 30 °C, similar to van der Werf et al. (2013). A $Q_{10}$ value of 1.5 implies a 50% increase for every 10 °C rise in temperature. Just like the moisture scalar, the abiotic scalar was standardized to a range from 0 to 1, and capped at a minimum of 0.1. Part of the turnover-exposed carbon is respired directly, based on a respiration fraction of 0.5. The remaining part degrades consecutively through the cwd (only originating from trees), fine litter and soil pools, and finally

enters the slow decomposition stage. Every degradation step is again subject to direct respiration. The belowground organic matter algorithm was simplified compared to GFED, because the belowground dynamics are not relevant for fire dynamics in our study area; fires do generally not occur in wetlands and peatlands in Africa. The LUE values and turnover rates used for the biomass and litter pools for each biome are summarized in Table 1. This table also gives the average effective turnover rates for the litter and cwd pools, after application of the abiotic scalar.

**2.2 Input datasets**

The model used MODIS (Moderate-resolution Imaging Spectroradiometer) Collection 6 satellite observation products with a 500-meter spatial resolution as input where available, and previous MODIS collections or coarser non-MODIS datasets otherwise (see Table 2). The meteorological input parameters were based on 0.25° resolution ERA-Interim reanalysis data (Dee et al., 2011) from the European Centre for Medium Range Weather Forecasts (ECMWF). The datasets used cover the

time period from 2002 to 2017, unless noted otherwise. The MODIS MCD15A2H product of fraction photosynthetically active radiation (fPAR; Myneni et al., 2015) was used in combination with reanalysis SSR (Dee et al., 2011) to calculate NPP (see Eq. 1). The distribution of biomass over tree and non-tree vegetation classes was based on the MODIS MOD44B



vegetation continuous fields (VCF; Dimiceli et al., 2015) product for the fractions of tree cover (FTC) and non-tree vegetation cover (NTV). Fire extent was based on burned area (BA) from the MODIS MCD64A1 dataset (Giglio et al., 2018). The decomposition of litter was based on temperature and soil moisture scalars derived from ERA-Interim reanalysis air temperature (2-meter temperature) and soil moisture (volumetric soil water layer 1) data. The classification of biomes

was based on the MODIS MCD12Q1 land cover type product, collection 5.1 (Friedl et al., 2010). The last available year of data for this product, 2013, was also used for subsequent years. For this study, the Land Cover Type 2 classification scheme produced by the University of Maryland (UMD) was used.

The 500-meter resolution MODIS input datasets were spatially aggregated to 0.050°, 0.125°, and 0.250° resolution using

average-based aggregation. As an exception, the qualitative land cover type data was aggregated using majority-based aggregation, by assigning the most frequently occurring land cover class to the aggregate grid cell. The 0.25° resolution reanalysis data was resampled to 500-meter resolution by nearest-neighbour interpolation, i.e. by using the reanalysis 0.25° grid cell value nearest to each MODIS pixel. All MODIS data with sub-monthly temporal resolution were averaged to monthly resolution, using the number of days in the month as weights.

**2.3 Model optimization**

We tuned our model to match satellite-based data on aboveground woody biomass (AGBw) (Avitabile et al., 2016) and field measurements of fuel load and consumption (van Leeuwen et al., 2014). Since NPP is the driver for biomass growth, we first ensured that biome-level NPP corresponded to GFED4 (van der Werf et al., 2017). Then, the AGBw was optimized to agree with observation-based gridded estimates by Avitabile et al. (2016), by tuning the turnover rates per biome. As a first order

approximation we tuned AGBw with the stem turnover rate, since the stems of trees hold at least 95% of the total AGB for all forest biomes (Poorter et al., 2012). Herbaceous (i.e. non-tree) biomass is typically below 250 gC m$^{-2}$ and therefore within the uncertainty range of the dataset by Avitabile et al. (2016). After optimization of the stem biomass, the turnover rates of the leaf, grass and root pools were adjusted to attain root-stem-leaf biomass ratios (i.e. root-shoot ratios) in line with the biome-specific ratios as reported by Poorter et al. (2012). The previously described subdivision of root biomass into separate

coarse and fine root pools was used to improve root-shoot ratios. The chosen turnover rates also influence the amount of litter produced. Even though the amount of tree biomass is not always relevant for fires, since most African fires are ground fires and deforestation mechanisms are not specifically part of the model, it does determine the amount of cwd and part of the fine litter produced.

In the final validation step, modelled FL and FC were compared to the compilation of field measured values by van Leeuwen et al. (2014). For the African continent the database contained 16 measurement records that reported FL and FC, of which 9 are grouped into different fuel classes (e.g. grass, leaves, litter, cwd). Additional field studies compiled by Scholes et al. (2011) on FL in African savannas were included in the comparison, giving 73 measurements on total FL. For all field



records used, measured FL consists of grass, litter, cwd, and occasionally leaves. As a consequence, the total FL estimates reported in our comparison to the measurements involved a variable number of model pools, dependent on what fuel classes were measured in the corresponding field study. Our definition of FL does not include the stem fuel class (and thus is not the same as AGBL) as this class was not reported in any field record used, which all consider ground fires in grass-dominated

biomes where trees are generally not affected. The field measurements were collocated with model results based on the field plot coordinates. The modelled FL was optimized for each biome separately by tuning turnover rates of the grass and litter pools to match measured average FL and spread in measurements. The root-shoot ratios were not significantly affected by this parameter tuning. Due to the lack of sufficient field data on FC (16 records, at only 6 unique locations), we validated FC only using the average of all records over all biomes.

All field measurements in the database were taken in savanna-type biomes, and all except one (in Burkina Faso) were taken in Southeast Africa (south of 12° S and east of 23° E), resulting in the sample set being less representative of other biomes and regions in Africa. Furthermore, the majority of records did not report separate measurements of specific fuel classes, and thus only provided an overall fuel load value for the combined fuel classes. As a result, the model validation was restricted to

a comparison of total FL and FC. The validation of individual fuel classes was also complicated by the large spatial variability in biomass allocation to fuel classes for field plots with similar properties, and because field conditions that determine the allocation ratios were unknown for most records (such as last fire occurrence, slash-and-burn or not, early or late season fires, etc.).

For most field records, the field plot coordinates were given with a precision of two decimal degrees. This yields an uncertainty of about 1 km, which is larger than the model pixel size of 500 meter. Therefore, more accurate coordinates with four decimal degrees precision were hand-picked based on the field site descriptions using Google Earth. Where possible, homogeneously vegetated areas were picked to remove the influence of other land cover types at sub-500 meter scale. For some field records the coordinate of a settlement or city nearby the field plot was reported instead of the actual plot, in which

case again a neighboring pixel was chosen, or the actual field plot was retraced in the vicinity of the reported coordinates. Many of the reported measurements were conducted before our study period, in which case the first model year, 2002, was used. For studies where only the year of measurement was known, the month in the middle of the regional fire season of the pixel was used. Finally, if there were recent burned area and related drops in biomass in a pixel, indicative of influence of recent fires, a neighboring unburned pixel was chosen.

**2.4 Simulations**

We ran our model at 500-meter native MCD64A1 resolution for the 2002-2017 period, with a monthly temporal resolution. A 200-year spin-up was done based on the 2002-2006 climatology, in order to stabilize the model pools and match total carbon in- and outflow. Additional simulations were performed for the three aggregated resolutions (0.050°, 0.125°, 0.250°)



to study the effect of spatial resolution on modelled biomass, litter and emissions. We restricted our analysis to the African continent, in particular to the Northern Hemisphere Africa (NHAF) and Southern Hemisphere Africa (SHAF) regions as defined in GFED (van der Werf et al., 2006). These two regions contain the African continent south of 23° N latitude and will be referred to as sub-Saharan Africa.

## 2.5 Resolution comparison

We calculated the differences that occur in modelled AGBL and fire emissions due to running the model at different spatial resolutions. We considered two categories of differences: those that occur as part of the model calibration and those that occur as part of the model simulation. The model calibration is dependent on resolution because it includes parameter tuning to match model pixels with field measurements, which is subject to a representation error. This error exists due to the scale mismatch in comparing field measurements to model grid cell averages (Janjić et al., 2018). At coarser resolution, the error is larger and as a consequence the model calibration is more biased. This leads to different model results at different resolutions, due to resolution-dependent model settings. We will refer to the differences between aggregated and 500-meter resolution model results due to different model calibration as *calibration* differences. Besides calibration-related differences, differences in the model simulation result from the spatial aggregation of input datasets and the subsequent coarse computation of the model algorithm. We will refer to the related differences in model results between aggregated and 500-meter resolution as *simulation* differences. We define simulation difference as the difference that occurs when running identical models with the exact same calibration, but at different spatial resolutions.

### 2.5.1 Calibration differences

We studied calibration differences by comparing the model calibrated at 500-meter resolution with an additional calibration at 0.25° resolution (Table 1, in parentheses). The 0.25° resolution calibrated model was also compared directly to GFED4 as it is based on a similar coarse resolution calibration, to determine whether our model simplifications were justified. This also allowed GFED4 to serve as an indirect reference to validate modelled emissions where FC field measurements were lacking. For the comparison to GFED4, the database without small fires was used (GFED4 instead of GFED4s), in order to compare the models using the same amount of burned area. The discrepancy between the burned area from GFED4 (without small fires but based on MCD64A1 Collection 5.1) and MCD64A1 Collection 6 was accounted for by raising GFED4 emissions according to the fraction of additional burned area in MCD64A1 Collection 6 compared to Collection 5.1.

### 2.5.2 Simulation differences

Besides studying calibration differences, we additionally quantified simulation differences as a result of running the model at different spatial resolutions. For this analysis we used the parameters based on the calibration at 500-meter resolution, to have the best model-data comparison. Using the same calibration, absolute and relative differences in simulation were




calculated as the coarse resolution results minus those of the 500-meter native resolution results. Beforehand, the 500-meter results were aggregated to the coarse resolution to be compared with, using average-based aggregation. A positive simulation difference indicates higher estimates at coarser resolution, and a negative difference indicates lower estimates at coarser resolution. The analyses were limited to the 2002-2017 annual average spatial fields, to focus on spatial resolution effects.

In order to understand the error mechanisms that lead to simulation differences and to quantify their contributions, we performed additional simulations with altered model algorithms and compared these to the base model simulation. For example, the contribution of fire in the overall simulation difference, and its contribution to error at coarser resolution, could be quantified by comparing an altered simulation without fire processes (i.e. BA = 0) to the base simulation with fires. This

contribution was calculated as the relative simulation difference with fires minus the difference without fires (altered minus base). Similarly, we compared simulations with and without fire-induced tree mortality, to study the contribution of that process in the overall simulation difference.

Notably, this method could only be used to quantify relative differences and not absolute differences, because the altered

simulations resulted in different model results, making absolute differences incomparable. In order to enable unbiased subtraction of relative differences, we calculated the log relative difference as:

$$\text{log relative difference} = \ln (X/X_{ref}) , \tag{6}$$

where, in our case, $X$ is the coarse simulation result and $X_{ref}$ is the 500-meter simulation result. Törnqvist et al. (1985) proposed this method as a replacement for the ordinary relative difference calculated as $(X - X_{ref})/X_{ref}$, because of its

additive, symmetric and normed properties. For positive values, the ordinary relative difference ranges from -1 to infinity, which is asymmetric and results in a positive bias when performing addition or subtraction. Using the log difference, we could quantify the isolated contribution of a process, by subtracting the log relative simulation difference of the altered simulation from that of the base simulation, without introducing bias. The log relative difference (i.e. $\log x$) approximates the ordinary relative difference (i.e. $y = x - 1$) for small values, but deviates strongly for large values due to the non-linear

scale, which has to be considered when interpreting the results. For example, an ordinary relative difference of 0.5 is equivalent to a log relative difference of 0.41, and analogously 1.0 translates to 0.69 log relative difference.

Using the same method, we also isolated the error that originates from the use of biome-specific LUE's and turnover rates. In the base model simulation, the most commonly occurring land cover class was used for the entire aggregate grid cell, and the

turnover rates and LUE of the majority biome were then applied to that grid cell. This leads to misclassification of the minority land cover classes (Foody, 2002), which we will refer to as *biome misclassification*. Furthermore, the biome-specific parameters of the majority biome were used without considering the minority biomes in the grid cell, leading to what we refer to as the *biome-specific parameter error*. We could account for this error by running the model for each biome





separately, so that the biome-specific parameters were correctly used for each individual biome. Then, the overall result was computed by summing the individual biome results, weighted by their respective fractional cover in the grid cell. Again, the altered run was compared to the base model simulation, in order to quantify the contribution of the error mechanism.

In a second 'per-biome' approach, we additionally treated all average-based aggregated input data on a per-biome basis. Because our model algorithm is nonlinear, Jensen's inequality exists between averaging the model result afterwards, as for the 500-meter model resolution, and averaging the input data (i.e. FTC, NTV, fPAR and BA) beforehand, as for the aggregated resolution simulations. We will refer to the error due to average-based aggregation of input datasets as the *input aggregation error*. In order to account for this error, we aggregated all MODIS 500-meter resolution input datasets to coarser

resolution according to the individual biome fractions in each grid cell. In other words, we created average-based aggregated input datasets for each biome area separately, instead of one aggregate for the entire grid cell area. These aggregation products were then used in the corresponding simulation of the individual biome, and the individual results were again summed afterwards. This altered simulation is only meaningful when the LUE and turnover rates are biome-specific as well, and should thus be seen as an addition to the altered simulation that accounts for the biome-specific parameter error, as

described in the previous paragraph.

Finally, a simulation was performed with the incorporation of a modified burned fraction (MBF), as described by van der Werf et al. (2017) and used in GFED4. They introduced the MBF to account for the underestimation of emissions in frequently burning areas at 0.25° model resolutions. The uniform burning of a fraction of an aggregated grid cell leads to

underestimation of emissions when fires occur in the subsequent months, because in this case fuel in the whole grid cell is lowered by the fires burning in previous months, also in areas that did not burn. In reality, the fuel is only lowered in the fraction of the grid cell that actually burned, and subsequent fires burn the sub-grid cell area that did not burn yet. The extent of underestimation is mainly dependent on the fire return time. Our 500-meter resolution model allowed us to directly test the effectiveness of implementing an MBF at coarser resolutions. We used a 4-month time period per burning season,

analogous to van der Werf et al. (2017).

## 3. Results

We ran our model for a range of spatial resolutions, based on model calibrations for either 500-meter or 0.25° resolution, to better understand the calibration and simulation differences. First, we discuss the results of the model calibration and validation for AGBw, FL and FC at 500-meter resolution. Next, we discuss the resulting AGBL and emission estimates

based on this model. Then we compare this to the results for the 0.25° resolution model calibration and relate this to GFED4. The remainder of this chapter is dedicated to simulation differences, i.e. the differences that occur between simulations with different spatial resolutions. For the study of simulation differences, we used the model calibrated at 500-meter resolution. In



this section we mainly focus on the differences between the 0.25° and 500-meter resolutions, and finish with the results for the intermediate resolutions of 0.125° and 0.050°.

## 3.1 Woody biomass and fuel load

The modelled total AGBw for Africa was 40.2 Pg C, compared to 40.6 Pg C for the Avitabile et al. (2016) reference dataset.
Area averaged values corresponded well, and the model was able to capture most of the spread in values (Fig. 1). This was as expected, since the model was calibrated to match the reference dataset. On a biome level, AGBw for tropical forest was 31.2 Pg C, which was 1.4 Pg C lower than the reference dataset. For woody and open savannas AGBw was 6.9 Pg C and 1.1 Pg C, respectively. Open savannas were slightly overestimated by 0.1 Pg C. The tropical forest, woody savanna and open savanna biomes contained the vast majority of tree biomass in Africa. Even though area averaged AGBw for other forest
types was significant, these biomes together constituted only 0.6% of African land surface and did not contribute significantly to total AGBw.

The comparison between modelled FL and field measurements based on the 500-meter model calibration is shown in Fig. 2a. A robust agreement was found, with an r value of 0.78 ($r^2$ = 0.60). The model tended to overestimate low FL and
underestimate high FL. Overall, the average, median and spread over all field sites agreed well between modelled and measured values (Fig. 2b). On average, FL for woody savanna and grassland was overestimated by approximately 15% (46 and 28 g C m$^{-2}$, respectively), whereas for shrubland values were underestimated by 11% (20 g C m$^{-2}$). Model estimates for open savannas agreed well with measurements, even though the range of values was underestimated for this biome. The shrubland and grassland statistics were both based on only 4 or 5 field measurements, which explains the large differences in
quantiles, and restricted the analysis to a comparison of averages and ranges.

Figure 2c and d shows the same comparison to field measurements, but simulated at 0.25° resolution. As in Fig. 2a and b, this comparison was based on the 500-meter resolution calibration, and thus only differed in simulation resolution. Compared to the 500-meter resolution simulation, FL modelled at 0.25° resolution had a much lower range and was
substantially underestimated in most biomes except grasslands. The flat regression slope indicates that the spread in measured FL was not captured at coarse resolution. This was partly because several FL measurements were within one 0.25° grid cell and thus yielded the same model value. The 0.25° simulation showed large estimation errors, especially for high FL measurements. One woody savanna measurement was incorrectly classified as tropical forest, leading to a large overestimation of FL (Fig. 2d, black triangles).



### 3.2 Fire emissions

The 2002-2017 annually averaged total fire emissions for sub-Saharan Africa were 0.68 Pg C yr$^{-1}$ based on the 500 m model, with an average FC of 249 g C m$^{-2}$ burned (Fig. 3, solid blue). The spatial distribution of emissions was dictated by burned area (Fig. 4a). The majority of emissions occurred in the subtropical savanna regions. About 90% of the fire emissions (0.61 Pg C yr$^{-1}$) originated from the woody and open savanna regions, where the majority (87%) of the annually averaged burned area was found. The highest FC was found in the tropical forest, where the average was 998 g C m$^{-2}$ burned. However, the burned area was relatively low in this biome, resulting in low emissions.

These 500-meter emission estimates for sub-Saharan Africa were 24% lower than GFED4 (without small fires), which estimated 0.90 Pg C yr$^{-1}$, and an average FC of 331 g C m$^{-2}$ burned (Fig. 3). All biomes with substantial emissions contributed to this difference, but woody savannas were the biggest contributor. The lower emissions in our model were the direct result of differences in FC compared to GFED4, because the amount of BA of the two estimates was identical. The spatial distribution of FC was less variable for most biomes in our model, except for the forest-dominated biomes. The variability of FC across biomes in the model was represented in a similar way as GFED4.

### 3.3 Calibration differences due to spatial resolution

The parameters used for the 0.25° resolution model calibration differed from the 500-meter calibration in terms of slower turnover rates for the stem, grass and litter pools for some biomes (Table 1). This resulted in roughly a 2.5 Pg C increase in AGBw and a 3.0 Pg C increase in AGBL. The majority of this increase was accounted for by the savanna biomes (both open and woody). Comparatively, non-woody AGB and litter increased more than woody AGB. Figure 5 shows the resulting modelled FL compared to field measurements – equivalent to Fig. 2, but for the 0.25° instead of 500-meter resolution model calibration. For this coarse calibration, again simulations for both 500-meter (panel a and b) and 0.25° resolution (panel c and d) are shown.

By calibrating the model at 0.25° resolution, the FL simulated at 0.25° resolution agreed better with measurements for all biomes (compare Fig. 2c, d and Fig. 5c, d). On the other hand, with this coarse calibration, the 500-meter resolution simulation significantly overestimated FL, and performed poorer than the 0.25° resolution simulation in terms of biome average and distribution (Fig. 5a and b). An exception was the shrubland biome, for which all 0.25° model pixels respective to the field sites were strongly influenced by low-biomass areas in that pixel. This resulted in a consistent underestimation of FL for both calibration resolutions. For the 0.25° calibration, the 500-meter resolution simulation still showed much better correlation with measurements (r$^2$ = 0.57), compared to the coarser simulation (r$^2$ = 0.07). The 0.25° calibration led to a regression slope that was steeper and closer to 1 for both resolutions, as high FL was amplified relative to low FL.





The increase in biomass and litter when using the 0.25° calibration led to higher emissions; 0.82 Pg C for both resolutions (Fig. 3, transparent bars). As expected, this was much closer to the estimate of 0.90 Pg C from GFED4 (without small fires) than for the 500-meter calibrated model. The largest change in emissions was in the open and woody savanna biomes. The resemblance to GFED4 emissions suggests that our simplified model is able to roughly reproduce GFED4 when calibrated at

the same resolution of 0.25°, while additionally enabling 500-meter resolution modelling.

### 3.4 Simulation differences due to spatial resolution

Besides calibration differences, model results varied because of resolution differences during the actual simulations. We calculated simulation differences between 500-meter and 0.25° resolution runs, using the model calibrated for 500-meter resolution (Fig. 6). For the 0.25° resolution simulation, AGBw and AGBL estimates were 4.0 Pg C and 4.6 Pg C lower than

the 500-meter simulation, respectively, with contributions from all biomes (Fig. 7a). The main positive differences were found at the transitions from barren to vegetated landscapes (e.g. at the fringes of the Sahara and Kalahari) and from land to water. Notably, smaller positive differences were also found in the southern part of West Africa. The average AGBL was lower at coarser resolution for all biomes, with a larger difference for biomes with more biomass (Fig. 7b).

Total fire emissions for the 0.25° run were 0.66 Pg C yr$^{-1}$, which was 3% lower than the 500-meter resolution simulation (Fig. 3, solid orange versus blue bars). Even though the total emission estimates at different resolutions were relatively similar, significant regional differences in emissions occurred, with deviations up to a factor of 1.5, and higher deviations (up to a factor of 4) at the border of water bodies and deserts (Fig. 6b). The lower emissions at 0.25 resolution were mostly the result of lower emissions in savannas and other grass-dominated biomes, whereas for tropical forests (i.e. Congo Basin) and

other forests, emissions were higher (see also Fig. 7a and b). Note that the relative differences shown for emissions are the same as for FC, because the burned area is equal for both resolutions.

### 3.5 Disentangling of mechanisms

The results in section 3.4 indicated that simulation differences were modest at continental scale, but substantial at regional scales. The various mechanisms that explain part of these differences were identified and quantified by doing additional

simulations with altered model configurations, and comparing them to the base model (see section 2.5.2).

### 3.5.1 Simulation differences for AGBL

Figure 8 shows the contributions of several error mechanisms to the total simulation difference in AGBL (shown in Fig. 6). Figure 8a and b depict the contribution of the biome-specific parameter error, and the input aggregation error, respectively. Figure 8c and d show the remaining simulation difference, after subtraction of these two error mechanisms. More

specifically, Fig. 8c shows the part related to fire processes (by doing a fire-off simulation), and Fig. 8d shows the





unexplained remainder. The effect of the MBF on AGBL was negligible and therefore not shown (but see Fig. 10). This is because frequent fires are mainly found in areas dominated by ground fires that only burn grass and litter, whereas AGBL is mostly determined by stem biomass and thus mainly affected by canopy fires, which are less frequent.

Errors due to biome-specific parameters played a very substantial role in the base model simulation difference in AGBL (Fig. 8a). For most of the African continent the majority of difference could be explained by this mechanism. The biome-specific parameter error accounted for lower AGBL for grass-dominated biomes and higher AGBL for forest-dominated biomes at coarser resolution. This shows that this error mechanism does not explain the strong negative difference in the tropics in the total difference (Fig. 6a). Differences were strongest at the transition borders of biomes, where the distribution

of land cover types is generally more heterogeneous. Examples are the transition of open savanna to woody savanna towards the equator (at 10° N and 15° S latitude), the transition of woody savanna to tropical forest towards the equator (at 5° N and 5° S latitude), and the transition towards the Sahara Desert.

Figure 8b shows the simulation differences in AGBL due to the input aggregation error. Compared to the biome-specific

parameter error, the input aggregation error was relevant in other areas, and the two error mechanisms partly neutralized each other (opposite signs in Fig. 8a and b). Substantial negative differences were found at the transitions to forest biomes (e.g. Congo rainforest, eastern South Africa, Madagascar). Positive differences were found at the transition to deserts and water bodies. Further investigation showed that these large positive differences occur where the majority-aggregated biome is water or desert, leading to large relative difference due to near-zero biomass. The remainder of simulation difference in

AGBL, after subtraction of the biome-specific parameter error and input aggregation error, was mixed positive and negative, of which all negative difference could be attributed to fire processes (Fig. 8c and d). This negative difference in the tropics explained a large part of the total simulation difference in that region (Fig. 8c, compare to Fig. 5b). This leaves an unexplained simulation difference of solely positive values, analogous to an overestimation of AGBL in the 0.25° resolution simulation.

Because the various error mechanisms influence each other, the order of isolation of different mechanisms affected the resulting relative difference. This was especially the case for the isolation of fire related error mechanism, for which the relative difference could vary by up to 25% difference dependent on the order of isolation. This was the case because the input aggregation error accounted for a part of the negative fire related difference in the woody savannas and the tropical

forest edge (note overlapping negative pattern in Fig. 8b and c), and the biome-specific parameter error accounted for a positive part in the open savannas. For the other mechanism, MBF, the order of isolation had a negligible impact.





### 3.5.2 Simulation differences for fire emissions

The results shown above concerned simulation differences in AGBL, which directly dictates the fuel load available for burning, and is thus a key precursor for fire emissions. The simulation differences in emission were generally less pronounced than for AGBL (Fig. 9). The biome-specific parameter error showed as a dipole of positive and negative
difference around biome transitions (Fig. 9a), similar to the pattern seen for AGBL. The difference due to the input aggregation error was mostly positive (Fig. 9b), and partly neutralized the biome-specific parameter error.

The isolated error related to the MBF was negative everywhere, and accounted for a substantial part of the total simulation difference in savanna regions (Fig. 9c). This shows that this measure is indeed able to remove errors at aggregated
resolutions related to short fire return times, as reasoned by van der Werf et al. (2017). The traditional way of accounting for fire in a model (unmodified burned fraction) causes an underestimation of emissions at aggregated resolutions in frequently burning landscapes, which translates to a negative simulation difference as shown in Fig. 9c. The remainder of simulation difference in emissions, after subtraction of all identified error mechanisms, was predominantly positive, especially in the region of the Congo tropical rainforest (Fig. 9d).

### 3.5.3 Relative contribution of error mechanisms

For all biomes, we identified the error mechanisms that explain the majority of total simulation difference in AGBL and emissions, except for tropical forest emissions (Fig. 10). From the -0.08 average relative difference in AGBL (blue dot in Fig. 10) for Africa between the base model running at 0.25° versus 500-meter resolution, 47% could be attributed to biome-specific parameter errors and an additional 2% to input aggregation errors. Furthermore, 16% was related to fire, 5% to the
MBF, and the remaining 30% was unexplained. This analysis was also performed for emissions, showing that from the -0.01 average relative difference (orange dot in Fig. 10) between the base model running at 0.25° versus 500-meter resolution, 30% could be attributed to biome-specific parameter errors and an additional 15% to input aggregation errors. The MBF accounted for 22%, and 33% remains unexplained.

On a biome level, most of the simulation differences were explained by biome-specific parameter errors. For AGBL, this mechanism explained the large majority of difference for the grass-dominated biomes. For tree-dominated biomes, input aggregation errors and fire processes were more important instead. For most biomes, various error mechanisms partly neutralized each other, resulting in a reduced overall difference. The MBF mostly affected emissions, and as expected the contribution was largest for biomes with considerable burned area and frequent fires. The unexplained remaining difference
was positive for all biomes, and only the emission differences for the grassland and cropland biomes were fully explained.





### 3.5.4 Simulation difference as a function of resolution

Across the four analyzed spatial resolutions (0.250°, 0.125°, 0.050°, 500-meter (which we assume to be 0.005° for this comparison)), the absolute differences in AGBL and emissions followed a gradual trend described by a natural logarithmic function. Exceptions were the difference in cropland AGBL, and in woody savanna emissions. The absence of a logarithmic

trend in these cases was caused by a mixture of pixels with positive and negative differences within the biome. This suggests that for these biomes the trend is explained by a combination of concave upward and downward logarithms for different pixels, as a result of variability within the biome. In case of emissions, the absence of a logarithmic trend for the whole of sub-Saharan Africa reflected the pattern of woody savannas. We estimated the sensitivity of the simulation difference as the derivative of the fit function. The general form of the fit function is:

$$a \log(x + b) + c \, ,  \tag{7}$$

where $x$ is the spatial resolution in degrees, and $a$, $b$, and $c$ are constants. Given the logarithm quotient rule:

$$a \log(x_1 + b) - a \log(x_2 + b) = a \log\left(\frac{x_1+b}{x_2+b}\right) , \tag{8}$$

and assuming $b \ll x$, each twofold increase or decrease in resolution results in a constant change in simulation difference. For example, the sensitivity for open savanna emissions is roughly $1.17 \cdot \log(2) = 0.81$ gC m$^{-2}$ per twofold change in

resolution (increase or decrease) (see Fig. 11b). This means that, independent of the initial resolution, a twofold finer (coarser) spatial resolution always leads to the same decrease (increase) in simulation difference. In other words, both at fine and coarse resolutions the model results are equally sensitive to resolution changes. Importantly, in cases where $b$ is not much smaller than $a$, the sensitivity decreases towards finer resolution (e.g. for tropical forest emissions).

### 4. Discussion

We estimated fire emissions of 0.68 Pg C yr$^{-1}$ for sub-Saharan Africa using an emission model based on native MODIS satellite spatial resolution of 500 meter. These relatively high-resolution estimates were compared to coarser resolutions, as used for most previous fire emission estimates such as from GFED and fire modules in DGVM's (Rabin et al., 2017). We analyzed differences due to spatial resolution occurring in both the calibration and simulation stage of our model.

With our simplified emission model, we were able to reproduce emissions from GFED4 on a continental and biome scale (Fig. 3). However, emissions for sub-Saharan Africa based on our 500-meter resolution model were 0.22 Pg C yr$^{-1}$ lower (-24%) than GFED4, with the largest difference for woody savannas. The difference with GFED4s emissions was larger because our model did not include small fire burned area. The emission estimates for our model calibrated at 0.25° resolution were 0.14 Pg C yr$^{-1}$ higher than for the 500-meter version, and more in line with GFED4 (-8%). Comparison of the 500-

meter and 0.25° resolution model calibrations illustrated that turnover rates governing biomass turnover and decomposition were required to be slower at aggregated resolution under the used calibration approach, which led to higher fuel load and

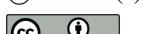



consequently higher emissions. This suggests that GFED likely overestimates fuel consumption due to its relatively coarse model resolution for similar reasons.

The lower fuel consumption estimates and underlying faster turnover rates for the 500-meter calibrated model version compared to the 0.25° resolution version, can partly be explained by a larger representation error when comparing model to field measurements. We showed that at 500-meter spatial resolution the representation error for model pixels compared to individual field measurements was greatly reduced, especially in heterogeneous landscapes with large spatial variation in biomass. The improved resolution additionally led to a larger sample of usable measurements, because multiple field plots that would otherwise be located in one 0.25° pixel could be compared individually. However, also at 500-meter resolution, part of the representation error remains, mostly because of the uncertainty in field plot location and time. Furthermore, the comparison to field measurements at finer resolution demands increased model complexity, since small-scale heterogeneity is no longer averaged out and thus has to be represented in the model.

When comparing our 0.25° calibrated resolution model version to GFED4, which runs at the same resolution, we found that the turnover rates governing biomass turnover and decomposition in our model were generally faster, despite the emission estimates being relatively smaller. This can partly be explained by the simplifications made in our model when compared to GFED4, such as the absence of herbivory, grazing, fuelwood collection and explicit deforestation mechanisms; all processes that remove additional biomass. Furthermore, because GFED4 is optimized globally and not only for Africa, turnover rates can be different for the same biome across continents. Lehmann et al. (2014) and Rogers et al. (2015) for example, discussed the differences in vegetation and fire characteristics among continents. This is also influenced by the availability of field data per continent, which is relatively poor for Africa. Finally, the faster turnover rates for the litter pools in particular can also be explained by the use of different soil moisture data and subsequent parameterization of litter decomposition in our model. Indeed, the effective turnover rates (i.e. after scaling by the abiotic scalar) for litter were slower and closer to GFED4 (Table 1). We have optimized our modelled AGBw to agree with the dataset developed by Avitabile et al. (2016) (see section 2.3). However, Bouvet et al. (2018) showed large negative biases in this dataset for savanna biomes, which suggests savanna stem turnover rates tend to be too fast in both our model and GFED4. This indicates that additional field data on biome-specific biomass and turnover rates is required to better evaluate our model.

Compared to the calibration difference in emissions of 0.14 Pg C yr$^{-1}$, the simulation difference in emissions of 0.02 Pg C yr$^{-1}$ was much smaller. Regionally however, simulation differences in AGBL and emissions were substantial (Fig. 6). At aggregated resolution, AGBL was lower almost everywhere and emissions were higher in the 10° N to 10° S belt but lower in the surrounding latitudes. In order to explain these differences, we identified at least three error mechanisms that can amplify or dampen each other: 1) biome-specific parameter error, 2) input aggregation error, and 3) temporal effects due to



aggregation of burned area fractions specifically, as spatial averaging affects sub-grid fire return time (MBF) and fuel build-up rates after a fire (post-fire fuel recovery, explained below).

## 4.1 Biome-specific parameter error

Overall, most of the simulation difference stemmed from the use of biome-specific parameters, especially where they varied considerably between biomes. In our model, the turnover rates were particularly variable, and they varied among biomes for all biomass pools. In contrast, in GFED4 only the stem pool turnover rates are set to biome-specific values, which we expect to result in relatively small errors. The biome-specific parameter error was largest in aggregate grid cells with a large sub-grid heterogeneity in land cover types and a large gradient in parameter values between neighboring pixels. Isolation of this error mechanism showed the largest AGBL differences in biome transition regions, where the variation in biomes is highest (Fig. 8a). Lower AGBL at 0.25° resolution for grass-dominated regions is explained by the misclassification of grid cell minority forest patches as grassland. At coarse resolution, average biomass is underestimated because the whole grid cell is simulated as a grassland (the majority biome), whereas at finer resolution the presence of a forest is revealed, with accompanying different turnover rates and LUE. Conversely, higher AGBL at 0.25° resolution for tree-dominated regions is explained by the misclassification of grid cell minority grassland as forest. This is comparable to previous studies that found an underestimation of small fragmented burned or forest area and an overestimation of large homogeneous burned or forest area, as a result of majority-based aggregation of binary classified pixels to coarser resolution (Eva and Lambin, 1998; Miettinen and Liew, 2009; Nelson et al., 2009; Silva et al., 2005).

The biome-specific parameter error in AGBL was mostly related to stem biomass, because AGBL mostly consisted of stem biomass, and because the stem pool had the largest range of turnover rates. The stem turnover rates for the open and woody savanna biomes differed by a factor of 7, which explains the notably large biome-specific parameter errors at transitions between those biomes, such as around the 10° N and 15° S latitudes (Fig. 8a). The large negative differences on the Eastern flank of the continent are also likely explained by a combination of the heterogenic mosaic of agriculture/savannas/forest biomes, and thus a large variability in tree cover and related turnover rates in this region. However, the error can similarly be significant for biomass pools other than the stem pool, since other turnover rates varied with up to a factor of 4 (see Table 1). This is especially important in biomes with little tree cover.

The biome-specific parameter error in AGBL directly affected emissions by determining the amount of fuel, and consequently again AGBL via the removal of fuel. However, for most pixels the relative differences in emissions were smaller than in AGBL (Fig. 9). Compared to 500-meter resolution, running the model at 0.25° resolution generally increased emissions in tropical forests and decreased emissions in savannas. A small area of grassland burning in an area predominately covered with forest results in an overestimation of emissions at aggregated resolutions, since the grid cell





average fuel load is mostly determined by forest biomass. The resulting emissions resemble a misclassified forest fire, instead of the actual grass fire. On the contrary, the burning of a small patch with high fuel load surrounded by a majority of low fuel load leads to an underestimation of emissions at coarse resolution. For these reasons, Fig. 9a shows positive-negative dipole patterns around the biome transitions. Emissions were overestimated on the more forested side of each biome

transition, and underestimated on the grassier side of each transition. These patterns were the direct result of the biome-specific parameter error in AGBL (Fig. 8a). There were no biome-specific parameters for fire, so no additional error was introduced in the calculation of emissions from AGBL. Notably, the relative error for emissions was much smaller than for AGBL. This can be explained by the fact that AGBL was mostly determined by stem biomass, whereas emissions were mostly determined by grass and litter (and leaf) biomass.

We expect that the AGBL after a fire is only minorly influenced by the biome-specific parameter error in emissions. Since most emissions originate from grass fires, there is a minor impact on stem biomass and thus AGBL. This is also indicated by the absence of an emission-like pattern in Fig. 9a, suggesting that this error in emissions is small where emissions are significant. By performing a simulation without fire induced tree mortality, we established that virtually all fire-related

resolution difference in AGBL (Fig. 8c) is caused by mortality, instead of by direct emissions. Fire mostly affects AGBL by killing trees, but this does not directly translate to emissions because there is a time lag in the combustion of dead stems (i.e. cwd) and the CC is relatively low.

## 4.2 Input aggregation error

The non-linear behavior of the model algorithm led to an additional input aggregation error because of Jensen's inequality.

This error was largest for input data with high spatial variability, and thus most apparent in grid cells with large heterogeneity in land cover types, where the bias due to averaging is strongest. Previously, we identified fire induced tree mortality to be the main reason for the fire-isolated resolution difference in AGBL (Fig. 8c). This pattern is clearly resembled in Fig. 8b, which suggests that mortality is strongly affected by the input aggregation error. This can be explained by the quadratic factor in calculating mortality, which amplifies Jensen's inequality.

By aggregating the input data for each biome separately, the input aggregation error was reduced by decreasing the variability related to the heterogeneity in land cover types. Using this method, the spatial resolution was effectively increased by a factor roughly equal to the number of biomes. However, variability inside individual biomes remains and is not accounted for using this approach, and likely accounts for the remaining unexplained simulation difference (Yuan et al.,

2007). Besides biomes, more or other aggregation classes can be chosen that ideally reduce the variability within those classes as much as possible with as little classes as possible. This could for example be a division based on tree cover intervals. However, the input aggregation error is unavoidable when modelling at aggregated resolutions, unless an appropriate estimator for Jensen's inequality can be derived to account for this error. Since the reanalysis climate datasets we




used had a 0.25° resolution for both our coarse and fine resolution simulations, no additional input aggregation error was introduced by these input datasets. However, in case finer resolution climate data is aggregated, Jensen's inequality will exist for these datasets as well. The error will probably be less substantial, as climate data is generally smoother and more homogenous spatially.

**4.3 Burned area aggregation (temporal effects)**

The aggregation of BA fractions to coarser resolution in particular led to additional errors, owing to temporal mechanisms. Firstly, the aggregation of BA altered fire return times, resulting in an underestimation of emissions at aggregated resolutions, especially in frequently burning areas. In GFED4, van der Werf et al. (2017) accounted for this effect by introducing the MBF. With our 500-meter resolution model, we were able to demonstrate the effectiveness of the MBF.

Indeed, the MBF (Fig. 9c) accounted for the majority of underestimation in emissions at aggregated resolution, especially in frequently burning areas (mostly savannas).

We introduce a second mechanism related to this, which occurs due to the temporal non-linearity of the modelled biomass buildup. This is a case of Jensen's inequality in the temporal dimension. The effect is illustrated in Fig. 12 for a hypothetical

case of stem biomass buildup. For this example, we assumed 100% CC and a uniform fuel load. At 500-meter resolution, the burned fraction is binary; a pixel is either completely burned or unaffected. In case the pixel burns, all stem biomass is removed by this hypothetical fire. In the next month, the biomass regrows from the start of the regrowth curve, where the slope is relatively steep, which leads to fast regrowth. On the contrary, in the case of aggregated burned fractions the same net amount of biomass is removed from the grid cell, but only a fraction of the total biomass. This results in slower regrowth

from a later point on the regrowth curve. This leads to an underestimation of fuel load at coarse resolution, due to slower biomass regrowth on average after a fire. Even though in normal model scenarios with partial CC the effect would be less extreme, it could still be of importance, especially in case of canopy fires. In general, the effect is stronger for slow turnover rates and short fire return times. In the case of a grass fire, most biomass has already recovered in the first months after the fire. Only in case of a short fire return time an effect could be noticeable. However, additional analysis is required to

quantify the contribution of this error mechanism.

**4.4 Small fires**

We found that emissions were in total lowered by $0.14 - 0.02 = 0.12$ Pg C yr$^{-1}$ (calibration minus simulation difference) when increasing model resolution from 0.25° to 500-meter resolution. For these results, we have relied solely on MCD64A1 burned area dataset which did not account for small fire burned area. Therefore, this difference may be offset by an increase

in emissions due to the inclusion of small fires. In GFED4s, emissions increased by 0.36 Pg C yr$^{-1}$ in our study region due to small fires, as compared to GFED4. An equivalent increase in emissions due to small fires in our model would offset the effects of spatial resolution threefold. However, our findings suggest that in a 500-meter resolution model the inclusion of



small fires may affect emissions differently. In GFED4s, small fire burned area is added to the MCD64A1 product burned area (Collection 5.1), followed by the calculation of emissions at 0.25° resolution as in GFED4. However, small fires mostly occur in croplands and at the border of tropical forests (i.e. deforestation), where the land cover is particularly heterogeneous (Randerson et al., 2012; van der Werf et al., 2017). Consequently, the GFED4s approach for calculating small fire emissions

is prone to the error mechanisms that occur at coarse resolution as described in this work. Our results suggest that small fire emission estimates, in particular, should be modelled at finer resolution or at least on a per-biome basis.

This is especially relevant considering the ongoing development towards finer resolution burned area products. For example, Hawbaker et al. (2017) published a 30-meter resolution burned area dataset for Northern America based on Landsat imagery.

Furthermore, Roteta et al. (2019) developed a dataset of 20-meter resolution burned area for Africa derived from the Sentinel-2 MSI sensor. Their preliminary product assessment indicated that a very substantial amount of burned area is still missed, even in GFED4s which includes small fires using a statistical approach. Our work illustrates how the development of finer resolution burned area datasets should be accompanied by the development of finer resolution emission models or better parameterization, in order to reduce errors. Even in our 500-meter resolution emission model, sub 500-meter

heterogeneity in burned area and fuel load is not accounted for, and could introduce additional errors.

We have shown that for relatively fine spatial resolution, the model was roughly equally sensitive to resolution changes as for coarser resolution (Fig. 11). The natural log relation implied that a twofold increase in resolution leads to a linear reduction of error. However, sub-500 meter modelling of fire emission could reveal new sources of error related to small-

scale heterogeneity. At these scales the log-relation might no longer be applicable. Dependent on the model precision demands, an optimal spatial resolution can be chosen for which the simulation difference becomes insignificant. However, the calibration difference can still be substantial, dependent on the representation error. A study by Nelson et al. (2009), who looked at the effect of spatial resolution on forest inventories, concluded that there is an optimal resolution of around 300 meters at which the pixel size is slightly smaller than the forest patch size and the essential heterogenic characteristics of the

landscape are captured. In line with our findings, this suggests there is a similar optimal resolution for burned area, and other spatial data used in fire emission modelling, at which finer resolution does no longer significantly improve captured variability in the data used, and computational resources are optimized.

## 5. Conclusions

We have developed a carbon cycle model to estimate fire emissions for sub-Saharan Africa, using the native spatial

resolution of MODIS data (500 meter). A key objective was to compare fire emission estimates at 500-meter resolution with more often used coarser resolutions such as the 0.25° resolution of GFED4. We estimated total fire emissions for sub-



Saharan Africa of 0.68 PgC yr$^{-1}$ averaged over 2002-2017. This is 24% lower than the most recent estimates from GFED4 (without small fires).

The difference was mainly caused by reduced representation errors in model calibration at finer resolution, when tuning the model to match field measurements of fuel load. In addition, estimates were different dependent on the resolution of the model simulation. At a more local scale, these simulation differences were substantial, with differences up to a factor 4 in regions with large landscape heterogeneity, such as biome transition zones. The error mechanisms we identified as main contributors to these simulation differences are all the result of spatial aggregation of the datasets used, and the consequent coarse resolution model simulation. Spatial aggregation leads to a reduction of data variability, both in case of majority-
based aggregation of land cover types, and in case of average-based aggregation of all other, continuous, input data. The identified error mechanisms explained most of the simulation difference, and the remaining unexplained difference is most likely caused by the variability inside individual biomes, which was not accounted for in our method. This variability can only be fully accounted for by running the model at native resolution. Furthermore, temporal effects, such as differences in post-fire fuel recovery, may also explain part of the remaining difference. These temporal effects should be further
investigated.

As a next step, we plan to run our model for the globe to improve global emission estimates, with a focus on highly heterogeneous regions such as deforestation zones. Understanding the underlying mechanisms that create errors in coarse resolution models enables the development of error reduction measures. This knowledge can be used to improve the next
version of GFED. Our results indicate that fuel consumption in GFED may be overestimated, at least for Africa. Whether correcting for the resolution-dependent errors discussed in this work will lead to lower global emissions in the next version of GFED, depends on to what extent small fire burned area may offset the decline in emissions.

**Code and data availability**

Code and model results are available upon request. GFED4s data is publicly available at https://www.globalfiredata.org/.

**Author contribution**

The research was designed by Dave van Wees and Guido R. van der Werf, and performed by Dave van Wees under principal supervision by Guido R. van der Werf, The manuscript was written by Dave van Wees with contributions from Guido R. van der Werf.



**Competing interests**

The authors declare that they have no conflict of interest.

**Acknowledgements**

This work was supported by the Netherlands organization for scientific research (NWO). We would like to thank Ioannis

Bistinas for useful discussions.

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



**Table 1: Biome-level model parameter values for light-use efficiency (LUE; $\varepsilon$, unitless) and turnover rates (t, years) for the stem, leaf, grass, litter and coarse woody debris (cwd) pools. Two additional columns give the average effective turnover rates (t eff., years) for the litter and cwd pools, after scaling by the abiotic scalar. Turnover rates that were different for the 0.25° resolution model calibration are given in parentheses.**

| Biome | $\varepsilon$ | $t_{stem}$ | $t_{leaf}$ | $t_{grass}$ | $t_{litt}$ | $t_{litt}$ eff. | $t_{cwd}$ | $t_{cwd}$ eff. |
|---|---|---|---|---|---|---|---|---|
| Evergreen needleleaf | 0.284 | 60 | 2 | 0.5 | 0.5 | - | 4 | - |
| Evergreen broadleaf | 0.354 | 60 | 1 | 0.5 | 0.5 | 0.8 | 4 | 6.1 |
| Deciduous needleleaf | 0.280 | 60 | 2 | 0.5 | 0.5 | - | 4 | - |
| Deciduous Broadleaf | 0.255 | 35 (60) | 0.5 | 0.5 | 0.5 | 1.7 | 4 | - |
| Mixed forest | 0.283 | 35 | 1 | 0.5 | 0.5 | 0.8 | 4 | 6.3 |
| Closed shrubland | 0.299 | 30 | 0.5 | 0.3 | 0.2 | 1.2 | 4 | - |
| Open shrubland | 0.208 | 30 | 0.5 | 0.3 | 0.1 | 0.3 | 1 | 2.6 |
| Woody savanna | 0.280 | 35 (40) | 0.5 | 0.3 (0.5) | 0.15 (0.2) | 0.5 | 1 (2) | 3.5 |
| Open savanna | 0.208 | 5 (10) | 0.5 | 0.3 (0.5) | 0.2 | 0.4 | 1 | 2.0 |
| Grassland | 0.229 | 18 | 0.5 | 0.2 | 0.1 | 0.2 | 1 | 2.4 |
| Cropland | 0.242 | 15 | 0.5 | 0.2 | 0.1 | 0.2 | 1 | 2.1 |



**Table 2: Overview of model input datasets.**

| Variable | Product | Spatial res. | Temporal res. | Reference |
| --- | --- | --- | --- | --- |
| fPAR | MCD15A2H | 500 m | 8-daily | Myneni et al. (2015) |
| FTC, NTV | MOD44B | 250 m | Annual | Dimiceli et al. (2015) |
| BA | MCD64A1 | 500 m | Monthly | Giglio et al. (2018) |
| Land cover | MCD12Q1 | 500 m | Annual | Friedl et al. (2010) |
| Shortwave radiation | ERA-Interim | 0.25° | Monthly | Dee et al. (2011) |
| Air temperature | ERA-Interim | 0.25° | Monthly | Dee et al. (2011) |
| Soil moisture | ERA-Interim | 0.25° | Monthly | Dee et al. (2011) |



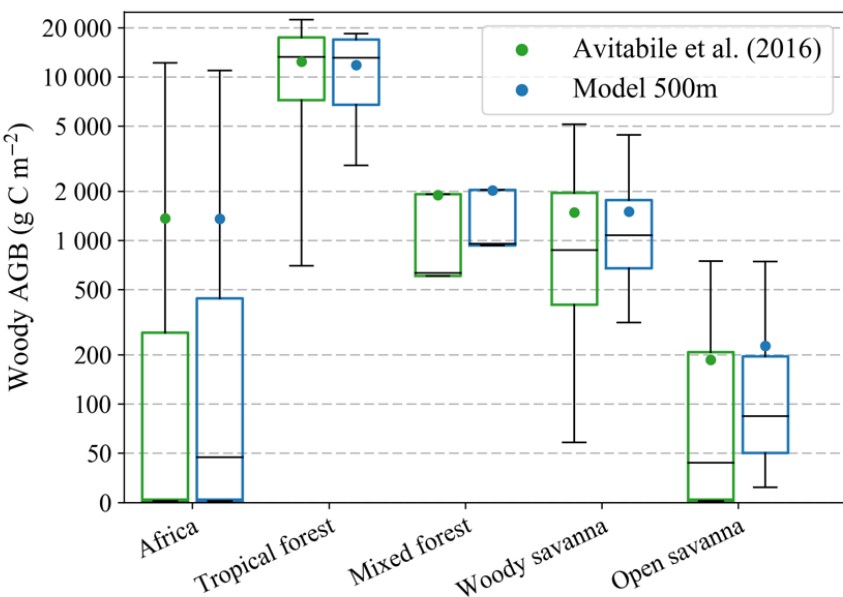

**Figure 1: Area-averaged woody aboveground biomass (AGBw) for sub-Saharan Africa as a whole, and for individual biomes with significant tree cover area. Both modelled values (500-meter) and those derived from the reference AGBw dataset by Avitabile et al. (2016) are shown. Boxplots show the mean (dots), median (horizontal line), 25th and 75th percentiles (boxes), and the 5th and 95th percentiles (whiskers). The 5th and 25th percentiles for Africa and the open savanna biome approach zero due to an abundance of pixels where AGBw is zero.**



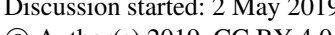



**Figure 2: Comparison of modelled FL and field measurements based on calibration for 500-meter resolution, shown as scatter plot and boxplots per biome for (a, b) 500-meter resolution and (c, d) 0.25° resolution simulations. Boxplots show the mean (colored dots), median (horizontal line), 25th and 75th percentiles (boxes), and the range of values (whiskers). The number of measurements involved is given below each boxplot.**





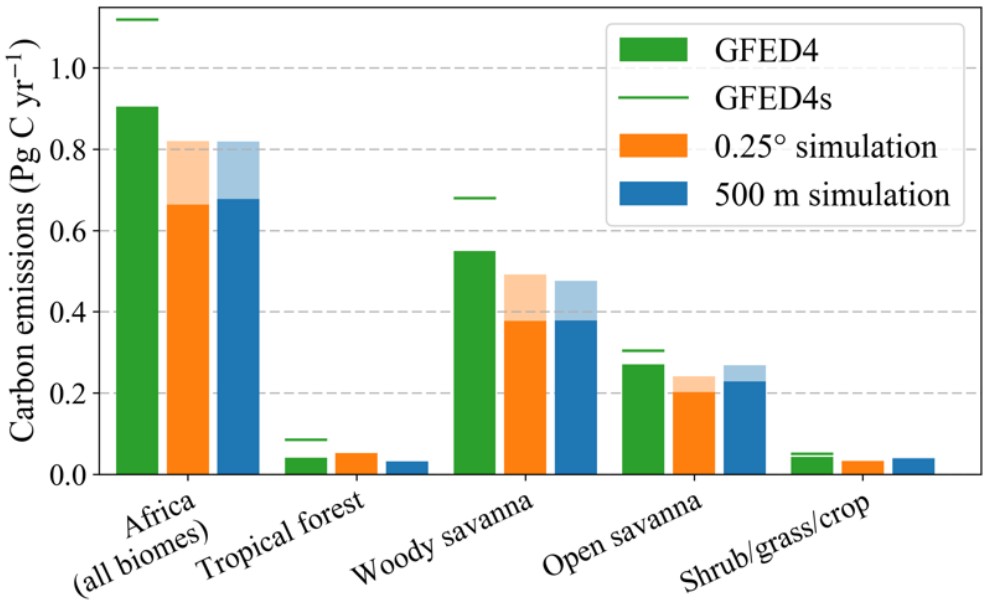

**Figure 3: Total fire emissions for sub-Saharan Africa and individual biomes, as compared to GFED4 (without small fires) and GFED4s (with small fires). Solid orange and blue bars show the estimates based on the 500-meter resolution model calibration. Transparent bars show the estimates based on the 0.25° resolution model calibration, highlighting that the calibration difference is larger than the simulation difference.**





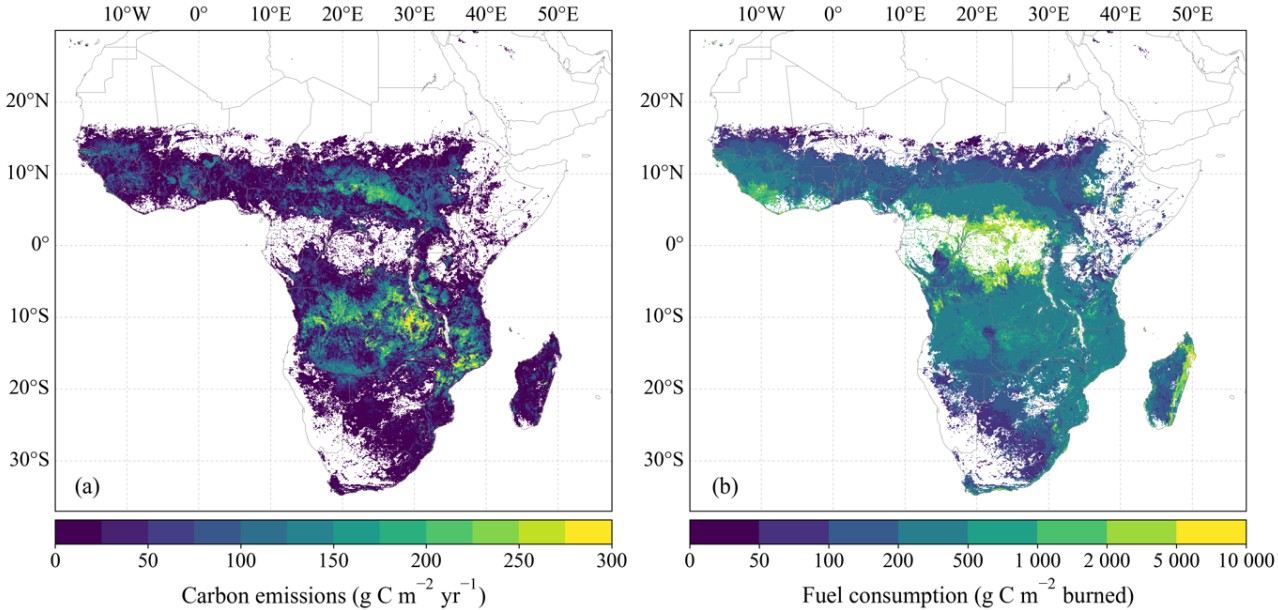

**Figure 4: Fire emissions (a) and FC (b) for the 500-meter resolution model, aggregated to 0.05° resolution for display.**

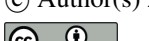



**Figure 5: Comparison of modelled FL at and field measurements based on calibration for 0.25° resolution, shown as scatter plot and boxplots per biome for (a, b) 500-meter resolution and (c, d) 0.25° resolution simulations. The Boxplot description is equivalent to Figure 2. Black triangles depict whiskers outside the plot range.**



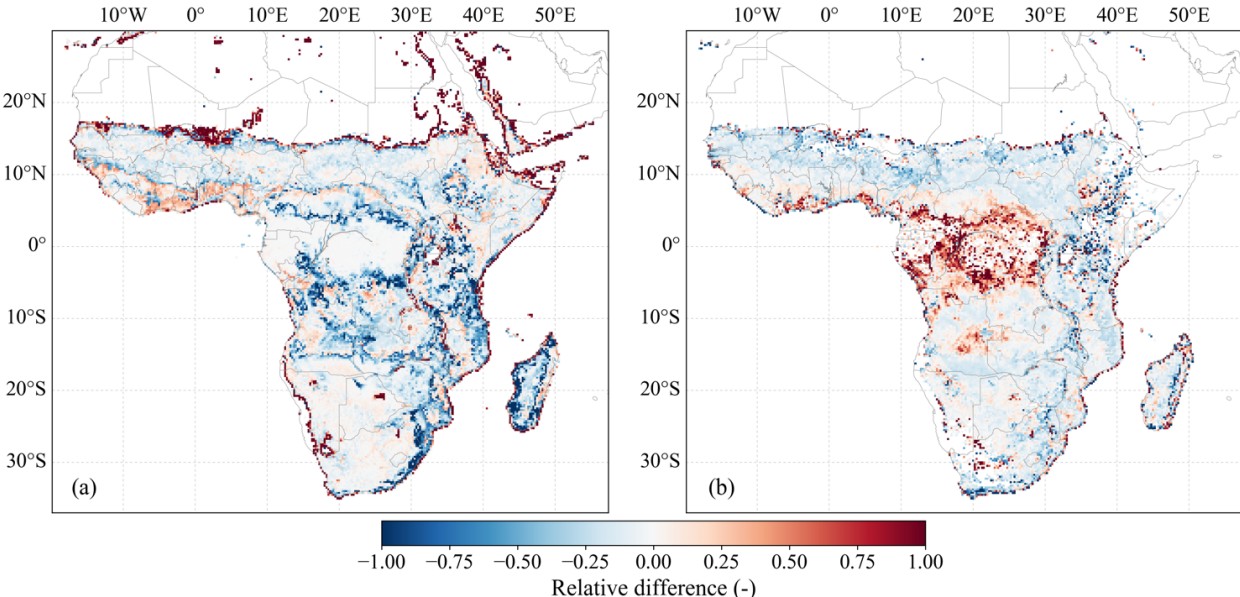

**Figure 6: Relative simulation difference as the natural logarithm of 0.25° over 500-meter resolution results, for (a) AGBL and (b) fire emissions (equivalent to FC). The 500-meter resolution model results are aggregated to 0.25° resolution beforehand. The relative difference in emissions is equivalent to the difference in FC. Positive values show 0.25° values higher than 500-meter values, and vice versa for negative values.**



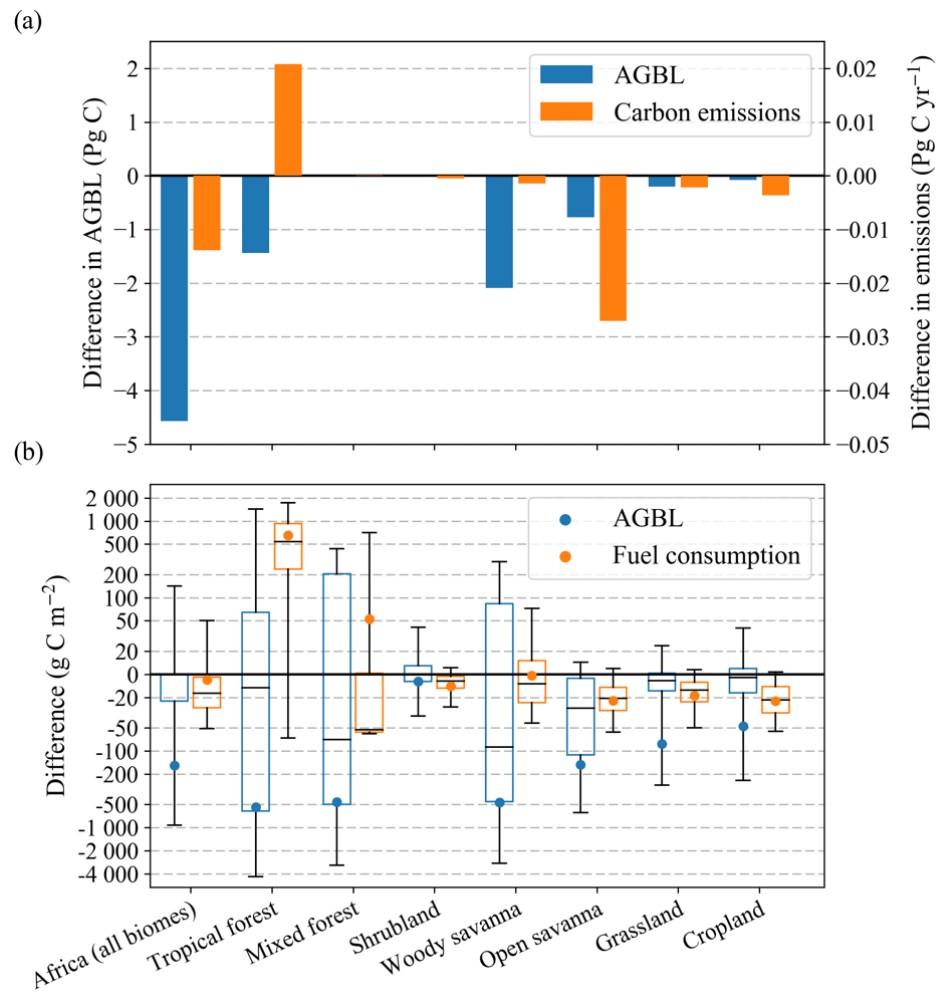

**Figure 7: Absolute simulation difference as 0.25° minus 500-meter resolution results for (a) total AGBL and fire emissions, and (b) area-average AGBL and FC (per area burned). Boxplots show the mean (colored dots), median (horizontal line), 25th and 75th percentiles (boxes), and the 5th and 95th percentiles (whiskers).**







**Figure 8: Relative simulation differences in AGBL for various isolated mechanisms, as the natural logarithm of 0.25° model result over 500-meter model result. (a) shows the isolated difference due to the biome-specific parameter error, (b) the additional difference due to the input aggregation error, (c) the remaining difference related to fire processes, and (d) the remaining unexplained difference after subtraction of (a), (b) and (c). The 500-meter resolution model results are aggregated to 0.25° resolution.**





**Figure 9: Relative simulation differences in fire emissions (and FC) for various isolated mechanisms, as the natural logarithm of 0.25° model result over 500-meter model result. (a) shows the isolated difference due to the biome-specific parameter error, (b) the additional difference due to the input aggregation error, (c) the difference accounted for by implementation of the modified burned fraction (MBF), and (d) the remaining unexplained difference after subtraction of (a), (b) and (c). The 500-meter resolution model results are aggregated to 0.25° resolution.**



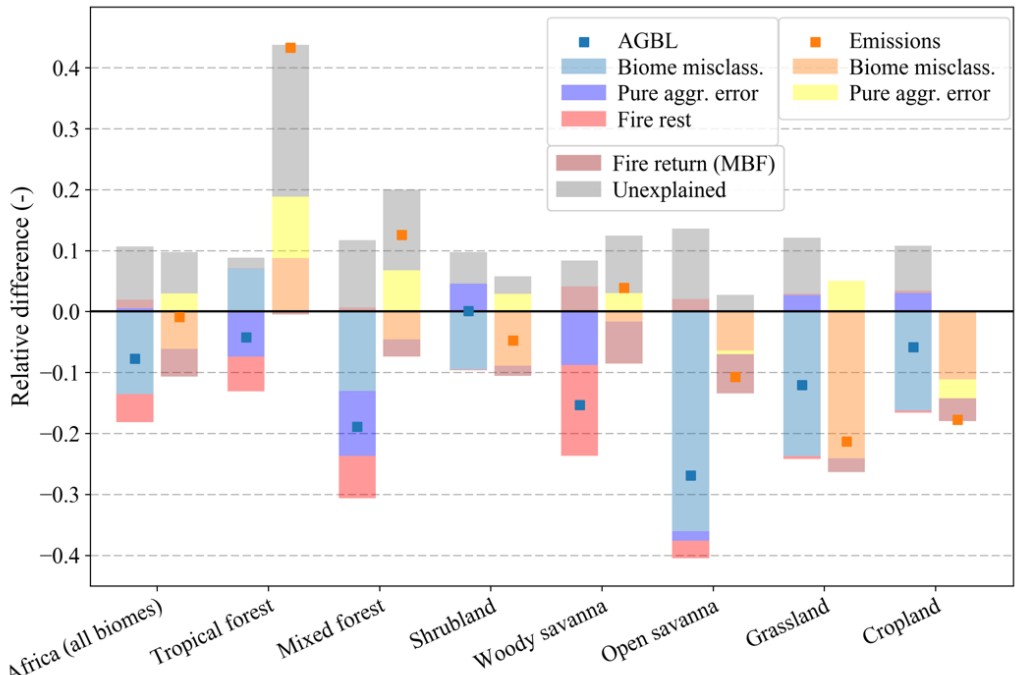

**Figure 10: Relative simulation difference in AGBL and fire emissions for sub-Saharan Africa and for individual biomes, as the natural logarithm of 0.25° model result over 500-meter model result. Stacked bars depict the contribution of various error mechanisms to the overall resolution difference in AGBL and fire emissions, by successive isolation of the biome-specific parameter error, the input aggregation error, the MBF and finally the remaining difference related to fire (Fire rest). Furthermore, the unexplained remainder after removal of all identified mechanisms is shown (Unexplained).**



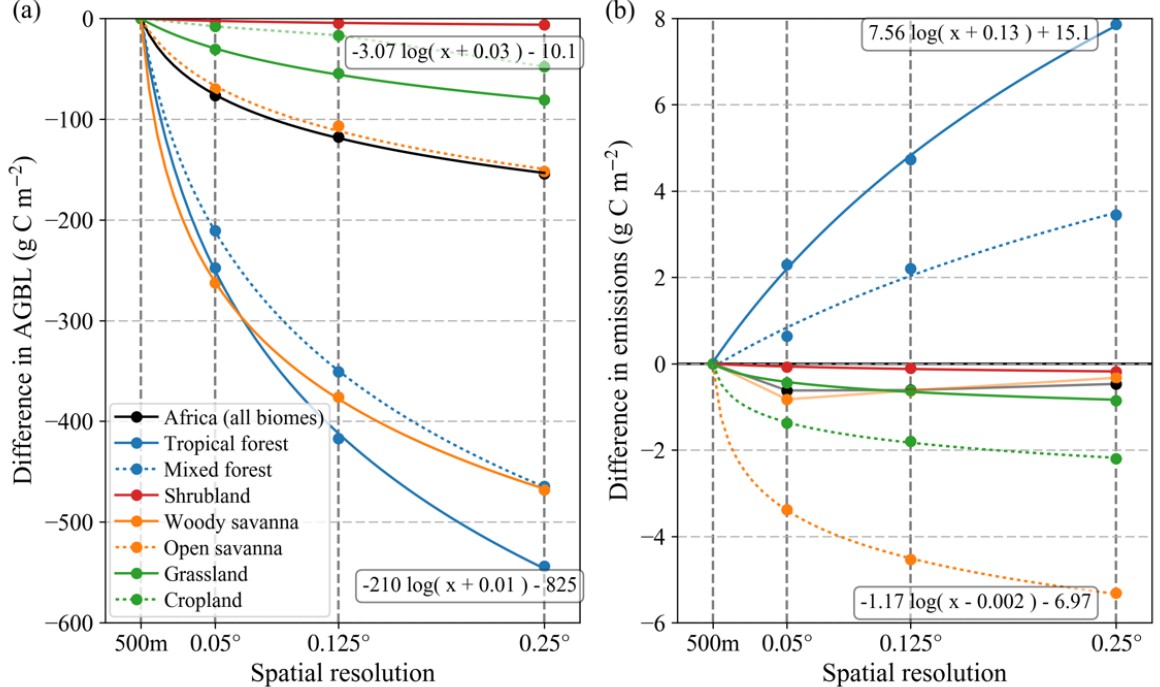

**Figure 11: Average absolute simulation difference per biome compared to 500-meter, for (a) AGBL and (b) fire emissions, versus spatial resolution. Data points are fitted with a natural logarithmic function where applicable. Inserted formulas show the corresponding fit functions for the two outermost lines. The plot axes are linear. For this plot, 500-meter resolution is assumed a resolution in degrees of 0.005°.**


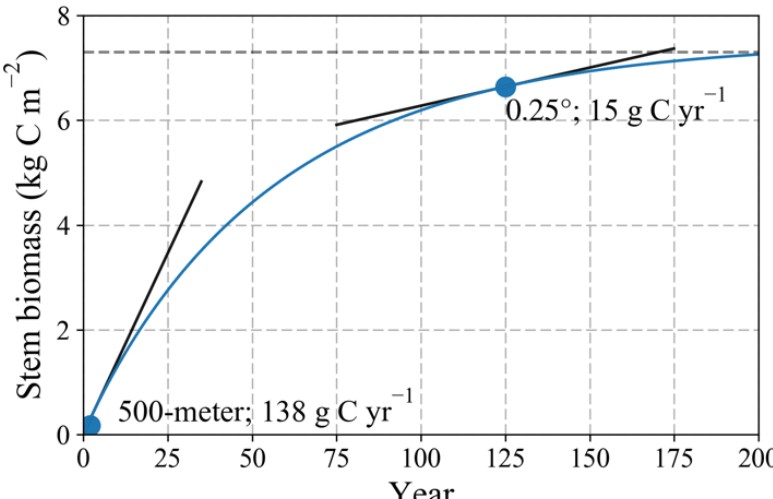

**Figure 12: Typical stem biomass growth curve.**