# Peer review of "Modelling African biomass burning emissions and the effect of spatial resolution"

_Geoscientific Model Development, 2019_

## Short Comment (SC1) · 17 May 2019

Dear authors,

In my role as Executive editor of GMD, I would like to bring to your attention our Editorial version 1.1:

http://www.geosci-model-dev.net/8/3487/2015/gmd-8-3487-2015.html

This highlights some requirements of papers published in GMD, which is also available on the GMD website in the 'Manuscript Types' section:

http://www.geoscientific-model-development.net/submission/manuscript_types.html

[Figure]

In particular, please note that for your paper, the following requirements have not been met in the Discussions paper:

- "The main paper must give the model name and version number (or other unique identifier) in the title."

- "If the model development relates to a single model then the model name and the version number must be included in the title of the paper. If the main intention of an article is to make a general (i.e. model independent) statement about the usefulness of a new development, but the usefulness is shown with the help of one specific model, the model name and version number must be stated in the title. The title could have a form such as, "Title outlining amazing generic advance: a case study with Model XXX (version Y)"."

- "All papers must include a section, at the end of the paper, entitled 'Code availability'. Here, either instructions for obtaining the code, or the reasons why the code is not available should be clearly stated. It is preferred for the code to be uploaded as a supplement or to be made available at a data repository with an associated DOI (digital object identifier) for the exact model version described in the paper. Alternatively, for established models, there may be an existing means of accessing the code through a particular system. In this case, there must exist a means of permanently accessing the precise model version described in the paper. In some cases, authors may prefer to put models on their own website, or to act as a point of contact for obtaining the code. Given the impermanence of websites and email addresses, this is not encouraged, and authors should consider improving the availability with a more permanent arrangement. After the paper is accepted the model archive should be updated to include a link to the GMD paper."

Thus include the models name and its version number in the title of your manuscript. Furthermore provide a reason (outside of your control) why your model is not publicly

available. If you are able to make it publicly available, please make the exact version, your article refers to, available via a permanent archive providing a DOI (e.g. Zenodo).

Yours,

Astrid Kerkweg

---

## Referee Comment (RC1) · João Silva (Referee) · 6 Aug 2019

General comments: This work assesses the effects of spatial resolution on the estimates of biomass burning in Africa, showing that the differences on estimates when modelling is performed at different resolutions is mainly due to parametrization errors and to input aggregation errors. The results are very relevant for the community dealing with pyrogenic emissions and for other areas where modelling is dependent on spatial inputs. The manuscript is very well structured and clear. Assumptions are clear and modelling is based on a simplified version of a very well established model (GFED).

Specific comments: In the Model Description section, model parametrization (e.g. lightuse efficiency, turnover rates) is defined according to biome-specific values. However, in the Input Datasets section it is said that "The classification of biomes was based on the MODIS MCD12Q1 land cover type product". A biome and a land cover type have different meanings. I understand and agree with the approach of using an input of land cover type with a relatively high spatial resolution, instead of biomes. I just suggest to acknowledge in this part of the manuscript the difference between biome and land cover type.

"Fire return times", used several times in the manuscript, should be replaced with "fire return interval", the terminology usually followed when describing fire regimes.

FireMIP should be defined and a reference added.

In the Fire Emissions section, it is mentioned that "The spatial distribution of emissions was dictated by burned area (Fig. 4a)." It would be very useful to add a map of burned area in Africa to figure 4.

---

## Referee Comment (RC2) · Anonymous Referee #2 · 28 Aug 2019

This paper aims to quantify the impact of resolution on a fire emission model based on MODIS burned area products. The authors distinguish and quantify three main sources of errors. The paper is very well written, generally clear and the conclusions are of great interest to the scientific community. The authors show a good grasp of the fire emission model, and of how errors can be propagated in non-linear models. The conclusion reached represent an important step towards a possible improvement of fire emissions, one of the key component for any air quality or atmospheric composition modelling enterprise. The paper can be published nearly as is as I have no general comments. A few remarks below:

[Figure]

• Page 2, line 4-6, if not too much trouble, a short comment to explain the difference between Africa's share of burned area (70%) and fire carbon emissions (roughly half) would be welcome. • Page 2, it would be desirable to link better paragraphs 2 and 3, maybe by harmonizing the terms used: is the "fuel load model" of paragraph 2 a component of the "fire emission model" of paragraph 3 (for the burned area approach)? This is clarified later at line 33, but maybe a clearer definition of "fire emission model" and of its components could be a good idea. • Page 4, line 25, maybe emphasize that the work focused on the burned area approach (accounting for small fires using active fire). Some of the results can certainly be used for fire estimates based on active fires (I am thinking of the biome related error for example), but probably not all of them. • It would certainly help the reader to have a geographical map showing the different biomes in Africa as used in the fire emission model. This could either be a new plots, or be shown together with Figure 6, 8 or 9 for example. • The impact of the meteorological input was not discussed, understandably since no consistent dataset at a resolution finer than 0.25° exist for the period considered. The ECMWF high resolution now proposes meteorological output at 0.1° resolution, so maybe for later work the impact of meteorological input could be considered as well.

―――――――――――――――

---

## Author Response (AR1)

**Reply to referee 1 (RC1), Joao Silva**

We thank Dr. Silva for his constructive comments; below we first repeat his comment and then show our response.

**Comment 1:**

In the Model Description section, model parametrization (e.g. light-use efficiency, turnover rates) is defined according to biome-specific values. However, in the Input Datasets section it is said that "The classification of biomes was based on the MODIS MCD12Q1 land cover type product". A biome and a land cover type have different meanings. I understand and agree with the approach of using an input of landcover type with a relatively high spatial resolution, instead of biomes. I just suggest to acknowledge in this part of the manuscript the difference between biome and landcover type.

**Response to comment 1:**

We changed:

Page 8, line 4-5: 'The classification of biomes was based on the MODIS MCD12Q1 land cover type product, collection 5.1 (Friedl et al., 2010).'

To:

Page 8, line 6-7: 'We delineated biomes in terms of land cover types based on the MODIS MCD12Q1 land cover type product, collection 5.1 (Friedl et al., 2010).'

This was done to stress that land cover types are used to demarcate biomes.

**Comment 2:**

"Fire return times", used several times in the manuscript, should be replaced with "fire return interval", the terminology usually followed when describing fire regimes.

**Response to comment 2:**

We agree and changed all occurrences of fire return time(s) to fire return interval(s).

**Comment 3:**

FireMIP should be defined and a reference added.

**Response to comment 3:**

Page 4, line 28: We have added the definition of the abbreviation 'FireMIP' and moved the reference to Rabin et al. (2017) following the abbrevation.

Comment 4:

In the Fire Emissions section, it is mentioned that "The spatial distribution of emissions was dictated by burned area (Fig. 4a)." It would be very useful to add a map of burned area in Africa to figure 4.

**Response to comment 4:**

For clarification we have added a map of average annual burned area in Africa to Figure 4.

**Reply to referee 2 (RC2), anonymous**

We thank the reviewer for a constructive review with helpful comments that strengthen the paper. Below we first repeat the comment and then show our response.

**Comment 1:**

Page 2, line 4-6, if not too much trouble, a short comment to explain the difference between Africa's share of burned area (70%) and fire carbon emissions (roughly half) would be welcome.

**Response to comment 1:**

**We have added to the sentence:**

Page 2, line 4-6: 'About 70% of global burned area occurs in Africa (Giglio et al., 2018), mostly due to surface fires with relatively low fuel consumption, leading to roughly half of the global fire carbon emissions (van der Werf et al., 2010).'

**Comment 2:**

Page 2, it would be desirable to link better paragraphs 2 and 3, maybe by harmonizing the terms used: is the "fuel load model" of paragraph 2 a component of the "fire emission model" of paragraph 3 (for the burned area approach)? This is clarified later at line 33, but maybe a clearer definition of "fire emission model" and of its components could be a good idea.

**Response to comment 2:**

We confirm that the 'biogeochemical model' or 'fuel load model' mentioned in paragraph 2 is indeed a component of the fire emission model in paragraph 3. To clarify this, we have changed Page 2, line 14-15 to:

'Two main satellite-based approaches to model fire emissions exist, based either on observed burned area in combination with a biogeochemical or fuel load model, ...'

**Comment 3:**

Page 4, line 25, maybe emphasize that the work focused on the burned area approach (accounting for small fires using active fire). Some of the results can certainly be used for fire estimates based on active fires (I am thinking of the biome related error for example), but probably not all of them.

**Response to comment 3:**

To clarify that we focused on the burned area approach we modified the sentence to: Page 4, line 29-30: 'To this end, we developed a fire emission model driven by burned area and capable of running at 500-meter spatial resolution, to produce a first emission estimate at this resolution for sub-Saharan Africa.'

**Comment 4:**

It would certainly help the reader to have a geographical map showing the different biomes in Africa as used in the fire emission model. This could either be a new plot, or be shown together with Figure 6, 8 or 9 for example.

**Response to comment 4:**

For clarification we have added a map of MODIS land cover types in Africa to Figure 4.

**Comment 5:**

The impact of the meteorological input was not discussed, understandably since no consistent dataset at a resolution finer than 0.25° exist for the period considered. The ECMWF high resolution now proposes meteorological output at 0.1° resolution, so maybe for later work the impact of meteorological input could be considered as well.

**Response to comment 5:**

[revised manuscript text omitted]
 delly grass mimory modulation correspondence hierarchy data for predicting delly grass mimory modulation correspondence hierarchy data for predicting delly grass mimory modulation correspondence hierarchy data for predicting delly grass mimory modulation correspondence hierarchy data for predicting delly grass mimory modulation correspondence hierarchy data for predicting delly grass mimory modulation correspondence hierarchy delly de

25 flux data for predicting daily gross primary production across biomes, Agric. For. Meteorol., 143(3-4), 189-207, doi:10.1016/J.AGRFORMET.2006.12.001, 2007.

Table 1: Biome-level model parameter values for light-use efficiency (LUE;  $\varepsilon$ , unitless) and turnover rates (t, years) for the stem, leaf, grass, litter and coarse woody debris (cwd) pools. Two additional columns give the average effective turnover rates (t eff., years) for the litter and cwd pools, after scaling by the abiotic scalar. Turnover rates that were different for the 0.25° resolution model calibration are given in parentheses.

| Biome                | ε     | t stem | t leaf | t grass | t litt | t litt eff. | t cwd | t cwd eff. |
|----------------------|-------|-------------------|-------------------|--------------------|-------------------|------------------------|------------------|-----------------------|
| Evergreen needleleaf | 0.284 | 60                | 2                 | 0.5                | 0.5               | -                      | 4                | -                     |
| Evergreen broadleaf  | 0.354 | 60                | 1                 | 0.5                | 0.5               | 0.8                    | 4                | 6.1                   |
| Deciduous needleleaf | 0.280 | 60                | 2                 | 0.5                | 0.5               | -                      | 4                | -                     |
| Deciduous Broadleaf  | 0.255 | 35 (60)           | 0.5               | 0.5                | 0.5               | 1.7                    | 4                | -                     |
| Mixed forest         | 0.283 | 35                | 1                 | 0.5                | 0.5               | 0.8                    | 4                | 6.3                   |
| Closed shrubland     | 0.299 | 30                | 0.5               | 0.3                | 0.2               | 1.2                    | 4                | -                     |
| Open shrubland       | 0.208 | 30                | 0.5               | 0.3                | 0.1               | 0.3                    | 1                | 2.6                   |
| Woody savanna        | 0.280 | 35 (40)           | 0.5               | 0.3 (0.5)          | 0.15 (0.2)        | 0.5                    | 1 (2)            | 3.5                   |
| Open savanna         | 0.208 | 5 (10)            | 0.5               | 0.3 (0.5)          | 0.2               | 0.4                    | 1                | 2.0                   |
| Grassland            | 0.229 | 18                | 0.5               | 0.2                | 0.1               | 0.2                    | 1                | 2.4                   |
| Cropland             | 0.242 | 15                | 0.5               | 0.2                | 0.1               | 0.2                    | 1                | 2.1                   |

**Table 2: Overview of model input datasets.**

| Variable            | Product     | Spatial res. | Temporal res. | Reference              |
|---------------------|-------------|--------------|---------------|------------------------|
| fPAR                | MCD15A2H    | 500 m        | 8-daily       | Myneni et al. (2015)   |
| FTC, NTV            | MOD44B      | 250 m        | Annual        | Dimiceli et al. (2015) |
| BA                  | MCD64A1     | 500 m        | Monthly       | Giglio et al. (2018)   |
| Land cover          | MCD12Q1     | 500 m        | Annual        | Friedl et al. (2010)   |
| Shortwave radiation | ERA-Interim | 0.25°        | Monthly       | Dee et al. (2011)      |
| Air temperature     | ERA-Interim | 0.25°        | Monthly       | Dee et al. (2011)      |
| Soil moisture       | ERA-Interim | 0.25°        | Monthly       | Dee et al. (2011)      |